# Activities against Lung Cancer of Biosynthesized Silver Nanoparticles: A Review

**DOI:** 10.3390/biomedicines11020389

**Published:** 2023-01-28

**Authors:** Jorge L. Mejía-Méndez, Edgar R. López-Mena, Eugenio Sánchez-Arreola

**Affiliations:** 1Laboratorio de Investigación Fitoquímica, Departamento de Ciencias Químico Biológicas, Universidad de las Américas Puebla, Ex Hacienda Sta. Catarina Mártir S/N, San Andrés Cholula 72810, Mexico; 2Tecnologico de Monterrey, Escuela de Ingeniería y Ciencias, Av. Gral. Ramón Corona No 2514, Colonia Nuevo México, Zapopan 45121, Mexico

**Keywords:** nanotechnology, nanomedicine, biological synthesis, silver nanoparticles, lung cancer

## Abstract

Nanomedicine is an interdisciplinary field where nanostructured objects are applied to treat or diagnose disease. Nanoparticles (NPs) are a special class of materials at nanometric scale that can be prepared from lipids, polymers, or noble metals through bottom-up approaches. Biological synthesis is a reliable, sustainable, and non-toxic bottom-up method that uses phytochemicals, microorganisms, and enzymes to induce the reduction of metal ions into NPs. Silver (Ag) NPs exhibit potent therapeutic properties that can be exploited to overcome the limitations of current treatment modalities for human health issues such as lung cancer (LC). Here, we review the preparation of AgNPs using biological synthesis and their application against LC using in vitro and in vivo models. An overview of the staging, diagnosis, genetic mutations, and treatment of LC, as well as its main subtypes, is presented. A summary of the reaction mechanisms of AgNPs using microbial cell cultures, plant extracts, phytochemicals, and amino acids is included. The use of capping agents in the biosynthesis of AgNPs with anticancer activity is also detailed. The history and biological activities of metal-based nanostructures synthesized with gold, copper, palladium, and platinum are considered. The possible anticancer mechanisms of AgNPs against LC models are covered. Our perspective about the future of AgNPs in LC treatment and nanomedicine is added.

## 1. Introduction

Nanotechnology is a fast-evolving field focused on producing structures to overcome agricultural, industrial, and therapeutic challenges. Objects found at the nanometric scale (1–100 nm) are known as nanomaterials (NMs) and are categorized into organic and inorganic. Organic NMs include liposomes, dendrimers, micelles, and biopolymeric nanoparticles (NPs). For biomedical purposes, organic NMs are design for drug or nucleic acid delivery, bone regeneration, wound healing, and imaging applications [1]. However, for cancer treatment, their implementation is limited, due to their low encapsulation efficacy, low mechanical strength, instability, and restricted cellular internalization [2,3].

In contrast to this category, inorganic NMs included quantum dots, carbon nanotubes, and metallic NPs. For therapeutic applications, NPs prepared from inorganic NMs are convenient, since their size (~10–500 nm), shape, high drug transport capacity, and high surface area to volume ratio, enable their use as adjuvants, biosensors, coatings, and drug carriers [4,5]. Metallic NPs are a unique class of nanostructures that are synthesized with noble metals such as silver, copper, gold, and platinum. 

In the past, silver was used to treat skin infections and maintain beverages in containers for long periods. At present, it is used to fabricate medical devices, treat dental problems, design domestic systems (e.g., air purifiers), innovate agricultural products (e.g., silver coatings to prevent infections in plants) [6], and formulate novel structures with applications in the treatment of chronic inflammation processes, infections caused by pathogenic microorganisms (e.g., *Proteus mirabilis*, *Candida albicans*, and hepatitis B virus), and carcinogenic events [7]. In nanomedicine and drug development, the advantage of AgNPs over other noble metal-based NPs relies on their intrinsic bioactive properties (e.g., antimicrobial, antioxidant, and cytotoxic), catalytic activity, chemical stability, and electrical conductivity [8]. In addition, AgNPs are advantageous nanostructures that can be fabricated with an adjustable size and shape at acceptable low costs and following standard protocols that do not require sophisticated laboratory materials [9,10]. 

AgNPs can be prepared following top-down (e.g., laser ablation, lithography, and vapor deposition) and bottom-up methods (e.g., flame spraying, microemulsion, and laser pyrolysis) [11]. Green synthesis, also known as biosynthesis, is a safe and cost-effective bottom-up technique that was first postulated in the 1990s by John Warner and Paul Anastas, as a plan to reduce environmental hazards [12]. Moreover, it was conceptualized as an way to reduce global, physical, and human health threats by minimizing the production of unnecessary wastes, overconsumption of energy, and cost-ineffective processes [13]. In non-biological methods (e.g., X-ray radiolysis and pulse radiolysis), silver salts such as silver perchlorate and silver sulfate are routinely used to synthesize silver-based nanostructures [14]. However, in biosynthesis, silver nitrate is predominantly used for its abundance and strong intrinsic bioactivities. For instance, it has been utilized to develop wound dressing systems, implants required in clinical treatment, medical catheters, dental modification composites, and innovative cancer therapy and diagnostic approaches [15,16,17]. 

Cancer is the term designated to a group of diseases caused by uncontrolled cell growth and proliferation. There are various risk factors that contribute to the incidence of this pathology, and they are categorized into intrinsic and non-intrinsic. Intrinsic risk factors are related to spontaneous mutations that occur randomly in DNA replication, whereas non-intrinsic risk factors include events that can be either modifiable (e.g., smoking, alcohol consumption, nutrient intake, and exposure to chemical carcinogens) or endogenous (e.g., genetic susceptibility, errors in DNA repair machinery, and dysregulated hormone levels) [18,19]. The distinct types of cancer are classified in accordance with their tissue or cell of origin and exhibit specific anatomic, histopathologic, molecular, genetic, and topographic features [20]. 

According to the Global Cancer Observatory (GLOBOCAN), the five types of cancer with the highest mortality worldwide are lung cancer (18%), colorectum cancer (9.4%), liver cancer (8.3%), stomach cancer (7.7%), and breast cancer (6.9%) [21,22]. Lung cancer (LC) is the leading cause of cancer related deaths worldwide; it is estimated that 236,740 persons were diagnosed with LC in 2022, whereas 130,180 will probably die because of this disease [23]. The treatment of LC and its subtypes, small cell lung cancer (SCLC) and non-small cell lung cancer (NSCLC), is based on chemotherapy, radiotherapy, immunotherapy, and targeted therapy regimens. However, despite their possible efficacy, their administration entails serious adverse effects, limited specificity, drug resistance development, and the possibility of relapse. Thus, the consideration of new and safe therapeutic alternatives is needed.

We consulted Google Scholar, PubMed, and Springer databases to present biosynthesized AgNPs as a safe, strong, and cost-effective alternative to the current treatment modalities of cancer, with a special emphasis on LC. The retrieved literature was consulted from 2012 to 2021 using the Elsevier search engine and considering the following keywords: green synthesis, biosynthesis, silver nanoparticles, and biofabrication. Using this strategy, 8.34% of articles reported the green synthesis of metal-based NPs, whereas 3.83% of the compiled literature merely focused on the production of AgNPs. Interestingly, while 13.03% of articles focused entirely on the cytotoxicity of green synthesized AgNPs, 39.78% reported their antioxidant activity. In the same regard, 5.29 and 21.8% of articles focused on the antimicrobial properties of biosynthesized AgNPs. On the other hand, the activity of biosynthesized AgNPs against LC or other types of cancer was reported indiscriminately among 69.03% of articles. In this review, an overview of risk factors, staging, genetic aberrations, diagnosis, and treatment schemes of LC and its subtypes is provided. The adverse effects of current treatment regimens against SCLC and NSCLC are emphasized. Evidence about the history and generalities of noble metals is compiled. The use of noble metals to produce NMs with anticancer applications is considered. The advantages and disadvantages of top-down and bottom-up methods are detailed, as well as characterization techniques. The use of reducing agents, and capping agents as biological sources to fabricate AgNPs against LC is reviewed. Moreover, the possible anticancer mechanisms of biosynthesized AgNPs in LC models (in vitro or in vivo) are examined.

## 2. An Overview of Lung Cancer and Its Subtypes 

LC, also known as bronchogenic carcinoma, is the leading cause of cancer-related deaths worldwide. In view of its histological and molecular characteristics, LC is classified into SCLC and NSCLC. The incidence of both subtypes is associated to various risk factors such as air pollution, age, family history, respiratory comorbidities (e.g., emphysema and chronic obstructive pulmonary disease), tobacco overconsumption, and occupational exposures [24]. In addition, infectious diseases caused by pathogenic microorganisms such as *Streptococcus pneumoniae*, *Chlamydia pneumoniae*, and human immunodeficiency virus (HIV) have been referenced as risk factors for LC development [25]. Regarding its epidemiology, LC is predominant in men in the United States [25], whereas in some countries of Latin America (e.g., Argentina, and Uruguay), its incidence has increased in both men and women over the last decade [26]. These facts can vary in accordance with various considerations such as geographic patterns, demographic variations, occupations, and the degree of exposure to potential risk factors [27]. As in other types of cancer, LC cells possess multiple genetic aberrations that stimulate their rapid proliferation, exacerbated metabolism (Warburg effect) [28], evasion of immune response, and overexpression of surface proteins (e.g., BCAT1, CD38, and PSTA1), which contribute to tumor formation and chemoresistance [29]. In addition, LC cells present genetic alterations in the regulators (e.g., cyclin-dependent kinases) of their cell cycle, which promotes the progression of LC [30]. The features of LC cells are depicted in Figure 1.

### 2.1. Small Cell Lung Cancer (SCLC)

SCLC arises from the central area and peripheric regions of the lungs [31]. It accounts for 15% of LC cases and is characterized by its aggressiveness, poor prognosis, low five-year survival rate (3.5%), and early metastasis to distinct organs, such as the brain and liver [32,33]. The tendency for relapse and development of drug resistance are other features that represent SCLC [34]. 

The staging of SCLC follows the tumor–node–metastasis (TNM) and the Veterans Administration Lung Study Group (VALSG) systems [35,36]. In the TNM classification, SCLC is categorized into limited and extensive. In the limited stage, the tumor is confined to one hemithorax and regional lymph nodes, which is opposite to the extensive stage, where the tumor exhibits distant metastases and is not confined to a single radiation field. Comparably, the VALSG system also stages SCLC into limited and extensive [37]. However, it considers that the disease may present local extension in the limited stage, whereas in the extensive stage, it can be manifested in contralateral lymph nodes, exhibit hematogenous metastases, and include malignant pericardial and pleural effusions [38]. Parameters that determine SCLC staging include laboratory tests, medical history, potential neurological symptoms, physical examination, and the diagnostic results of patient [39].

The diagnosis of SCLC is performed using computed tomography (CT), magnetic resonance imaging (MRI), and positron emission tomography (PET) [40]. For biopsy samples obtained from SCLC tumors, methods such as droplet digital polymerase chain reaction (PCR) and next generation sequencing (NGS) can be used to determine gene mutation profiles [41]. In SCLC tumors, the loss of Renitoblastoma 1 (*RB1*) gene expression has been correlated to 95% of tumors development, whereas mutations in the tumor protein *p53* gene have been reported in 65% of SCLC cases [42].To a lesser extent, the inactivation of the methyltransferase lysine methyltransferase 2D gene (KMT2D has been observed in 8% of SCLC tumors and 17% of SCLC cell lines (e.g., NCI-H2171) [43]. Other genetic alterations have been validated for amplified proto-oncogenes, such as MYC, MYCL, and MYCN; the relationship between these genes and the aggressiveness, progression, and treatment resistance of SCLC has been proposed in genetically engineered mouse models [44]. Alterations in genes such as members of the NOTCH family, and cell signaling pathways such as PI3K/AKT/mTOR, have been attributed to SCLC pathogenesis [45,46]. 

As in other types of cancer, SCLC is treated with various approaches, such as surgery, chemotherapy, radiation therapy, immunotherapy, and targeted therapy [47]. However, their use depends upon the stage of SCLC. During the limited stage, the preferred treatment for SCLC is chemotherapy and radiation therapy; chemotherapeutic agents such anthracyclines (e.g., doxorubicin), alkylating agents (e.g., cyclophosphamide), and platinum-based compounds (e.g., cisplatin and carboplatin) are administered to treat limited stage SCLC [48]. Moreover, if a proper response is observed after treatment with chemotherapy, radiation techniques such as prophylactic cranial irradiation (PCI) can be used to improve the progression free survival and reduce the incidence of symptomatic brain metastases in SCLC patients in the limited stage [49]. Alternative radiation techniques include elective nodal irradiation, intensity-modulated radiation therapy, and stereotactic ablative radiation therapy [50].

For the extensive stage, the administration of carboplatin, cisplatin, docetaxel, and topoisomerase inhibitors such as irinotecan has been document for the treatment of SCLC [51]. For patients who relapse after treatment with platinum-based chemotherapy (i.e., platinum-etoposide), the administration of other chemotherapeutic drugs such as temozolomide is indicated [52]. On the other hand, various drug combinations have been purposed as an strategy to treat SCLC in the extensive stage; some of these regiments include cisplatin and irinotecan, etoposide and cisplatin, and cyclophosphamide, doxorubicin, and vincristine [53]. Among the radiation therapies, patients with SCLC in the extensive stage can receive consolidation thoracic radiotherapy and PCI [54]. In addition, immunotherapeutic agents (e.g., atezolizumab, ipilimumab, and nivolumab) and targeted therapy agents (e.g., veliparib) have been tested to treat SCLC at this same stage [55,56].

### 2.2. Non-Small Cell Lung Cancer (NSCLC)

NSCLC tumors can originate from the upper and non-upper lobes of the lung [57]. It accounts for 85% of LC cases, possessed a complex genetic nature, and is categorized into further subtypes, such as adenocarcinoma (AC), large cell carcinoma, and squamous cell carcinoma (SqCC) [47,58,59]. The subtypes of NSCLC share biological features; however, they exhibit different genomic alterations, location within the lung, growth and progression patterns, and cellular origin [60]. For instance, SqCC is centrally located in the lung airways, exhibits a rapid growth-rate, and is associated with blood vessel invasion, bronchial obstruction, tumor cavitation, and fatal pulmonary hemorrhage cases [61]. In contrast, AC is related to the alveolar epithelium and bronchial mucosal glands, manifests multiple morphologic patterns (i.e., acinar, bronchioloalveolar, papillary, and solid), and exhibits various genetic alterations that can determine the effectiveness of therapeutic agents [62,63].

Several genetic aberrations contribute to the progression, drug resistance, and metastasis of NSCLC. For example, insertions or deletions in the epidermal growth factor receptor (*EGFR*) gene can promote its pro-oncogenic effects by contributing to the angiogenesis of tumors, inhibition of apoptosis, cellular proliferation, and metastasis of NSCLC [64]. On the other hand, alterations in genes such as the anaplastic lymphoma kinase (ALK) gene, *BRAF* gene, Kirsten rat sarcoma viral oncogene (*KRAS*), mesenchymal epithelial transition (*MET*) gene, and *ROS1* are frequently observed in patients with NSCLC-AC [65]. For patients with NSCLC-SqCC, mutations in the fibroblast growth factor receptor (FGFR1) gene, Erb-B2 Receptor Tyrosine Kinase 2 (*ErbB*) gene, and TP53 gene have been found [66].

In the TNM system, NSCLC is classified into three stages: local (IA, IB, and IIA), locally advanced (IIB, IIIA, and IIIB), and advanced (IIIB and IV) [67]. As in SCLC, NSCLC can be diagnosed using CT, MRI, and PET. In addition, it can be detected using chest x-ray (CXR), lung biopsy, and sputum analysis methods [68]. On the other hand, techniques such as endobronchial ultrasound (EBUS) and endoscopic ultrasound (EUS) can be used to examine NSCLC tumors located in suspicious mediastinal nodes [69].

The treatment of NSCLC is based on surgery, chemotherapy, immunotherapy, and targeted therapy. For patients with stage I- or II-NSCLC, surgical resection, cisplatin-based chemotherapy, and stereotactic ablative radiotherapy are recommended; however, their choice relies on the extension of the disease, tumor location, and patient viability for operation [70]. Similarly, stage III-NSCLC is treated with chemotherapy (e.g., paclitaxel and vinorelbine) or radiotherapy [69]. Moreover, platinum-based chemotherapy and immunotherapy with bevacizumab or pemetrexed have been indicated as first-line systemic treatments for patients with stage IV-NSCLC [71]. For this same stage, alectinib, crizotinib, and gefitinib are administered as targeted therapies against *ALK*, *ROS1*, and *EGFR*. In addition, for mutated genes such as *BRAF*, *KRAS*, and *MET*, dabrafenib, sotorasib, and tepotinib have been reviewed as targeted therapies [72].

As previously mentioned, LC treatment causes numerous adverse effects. In NSCLC treatment modalities, serious negative events have been documented; for instance, chemotherapy regimens can exert anemia, intestinal injury, and cardiotoxicity [73], whereas immunotherapy can cause concomitant pneumonitis and interstitial lung disease [74]. Alopecia, nausea, diarrhea, and pyrexia have been also observed during targeted therapy regimens [75].

## 3. A Brief History and General Features of Noble Metals with Biological Activity

Noble metals are high-density non-toxic agents characterized as distributed in nature and resistant to corrosion. Representative materials that belong to this category are silver, gold, copper, palladium, osmium, iridium, and rhodium [76]. In ancient times, many civilizations used these materials because of their intrinsic bioactivities; for instance, Greeks such as Hippocrates documented the use of silver foils to treat ulcers and wounds, whereas the Egyptians used copper to sterilize water. Comparably, silver nitrate was used by the Romans as an antibiotic with topical applications and as a material to manufacture urns [77]. On the other hand, gold compounds such as gold cyanide were investigated in 1890 by Dr. Robert Koch because of their capacity to be toxic to *Mycobacterium tuberculosis*, the etiological agent of tuberculosis [78,79]. Moreover, in 1920, gold salts such as auranofin were used to treat patients who had been diagnosed with lupus erythematosus, rheumatoid arthritis, and psoriatic arthritis [80]. General applications of metal-based nanomaterials are depicted in Figure 2.

Given the technological and scientific advances throughout human history, the photothermal, electrical, optical, and chemical features of noble metals were revealed, and they started to be implemented to synthesize chemotherapy medications such as cisplatin, as well as to produce NM products for the jewelry industry, dentistry, and oncology [81,82,83]. In nanomedicine, NMs prepared from noble metals such as gold are heavily used for biological imaging, molecular diagnostics, and photothermal cancer therapy [84]. Comparably, NPs synthesized with silver are used as carriers for proteins against breast, leukemia, and neuroblastoma cancer cells [85,86], or as radiosensitizers against in vivo glioma models [87,88]. Regarding other NMs, copper has been used to fabricate nanoclusters with applications in fluorescence imaging, MRI, and PET studies [89], whereas palladium based nanowires, nanoflowers, and nanosheets have been used as biosensors, wound healing agents, and photoacoustic imaging platforms, respectively [90]. Other applications of noble metal-based NPs are summarized in Table 1.

Since the dimensions of metal-based nanostructures can influence their capacity to target cells, cross physiological barriers, or carry therapeutic compounds to the desired sites of action, many methods have been developed to prepare them [110]. In top-down processes, bulk materials are transformed into smaller particles using methods such as laser ablation, flame spray pyrolysis, sputtering, and lithography [111]. To produce nanoplatforms with potential drug delivery and molecular imaging applications, top-down techniques are convenient, as they can offer structures with a consistent morphology, stability, and quality [112]. However, the use of these methods has many disadvantages, such as the generation of toxic chemicals, high heat generation, and low production yield [113]. Similarly, applications in clinical investigations of nanostructures fabricated through top-down techniques can be hampered due to particle contamination, which can lead to cellular toxicity and possible carcinogenicity [111,114].

In contrast, bottom-up methods induce the self-assembly of atoms, molecules, or their clusters into NMs; this phenomenon is driven by ionic bonding, van der Waals Forces, and water-mediated hydrogen bonding [115]. For the synthesis of metallic NPs, these methods are preferred, since they provide greater control over the size, shape, and chemical composition of the produced NPs. However, the removal of organic solvents, reagents, or toxic chemicals from the reaction mixture constitutes a major challenge [116]. Some bottom-up techniques used to produce metal-based NMs include biological methods, electrochemical procedures, chemical reduction processes, microwave-induced synthesis, and solvothermal methods [116]. 

As in other nanostructures, the physical and chemical features of metallic NPs are determined using spectroscopy and microscopy techniques. Concerning spectroscopy methods, Fourier transform infrared (FTIR) spectroscopy is utilized to assess the chemical composition of metallic NPs, whereas ultraviolet-visible (UV-vis) spectroscopy is implemented to study their agglomeration state and estimate their size [117]. In the same context, the formation, surface chemistry, and electronic properties of noble metal NPs can be analyzed using nuclear magnetic resonance (NMR) [118]. On the other hand, dynamic light scattering (DLS), also known as photo correlation spectroscopy, is implemented to study the hydrodynamic diameter, and surface charge (also known as ζ-potential) of noble metal NPs. To complement the structural and electronic information of noble metal-based NMs, X-ray absorption spectroscopy (XAS) can be useful [119]. 

On the other hand, microscopy methods such as transmission electron microscopy (TEM) and atomic force microscopy (AFM) are used to confirm the size and shape of metallic NPs [120], whereas scanning electron microscopy (SEM) is utilized to visualize their surfaces [121]. For the same purpose, dark-field microscopy is employed to investigate the physico-chemical composition of noble metal NPs, as well as their interaction with the surrounding environment [122]. 

## 4. Biological Sources for AgNPs Synthesis

AgNPs can be prepared following non-biological synthesis methods that involve physical and chemical processes. Common techniques that belong to this category include micropatterning, pyrolysis, thermolysis, nanolithography, and laser ablation. Originally, physical approaches were ideal for producing uniform metal-based nanostructures with applications in tissue engineering, cellular studies, and biosensors development [116]. However, their use can involve equipment with a high energy consumption, require long periods to achieve optimal operating temperatures, and may provide low product yields. In the same context, chemical methods (e.g., microemulsion and sol-gel) represent a fast and efficient approach to synthesizing nanostructures with the capacity to entrap and release therapeutic compounds using external or internal stimuli (e.g., light, pH, or temperature) [123,124]. However, their implementation often involves the use of hazardous organic solvents (e.g., hydrazine and dimethyl formamide), physical conditions that are challenging to achieve (e.g., vacuum pressure and high temperatures), and the final products can become easily destabilized under biological conditions [11,125].

In contrast, biological methods represent a cost-effective, simple, and environmentally friendly technique for producing AgNPs at high yields. In this approach, reducing agents are required to promote the precipitation of AgNO_3_ ions from salt solution and their consequent nucleation and growth into AgNPs [126,127]. Common reducing agents reported in the scientific literature include amino acids (e.g., tyrosine and tryptophane), microorganisms (e.g., algae, bacteria, fungi, and viruses), plant extracts (e.g., *Clitoria ternatea* and *Solanum nigrum*) [128], phytochemicals, and enzymes (e.g., acetyl xylan esterase and glucosidase) [129,130]. Examples of biological sources to biosynthesize AgNPs are represented in Figure 3.

Depending on the selected reducing agent, temperature, and pH, the functionality, size, shape, monodispersity, and ζ-potential of the formed AgNPs can vary and arise from distinct mechanisms [131]. The activity, size, shape, and ζ-potential of biosynthesized AgNPs against different cancer models is summarized in Table 2. The different sources for biosynthesizing AgNPs and their reaction mechanisms are described in the following paragraphs.

### 4.1. Microbial Cell Cultures

In microbial cell cultures such as bacteria and fungi, the formation of AgNPs can occur at an extracellular or intracellular level. In the extracellular process, metal ions are reduced to NPs using microbial cellular components such as cell wall components, enzymes (e.g., oxidoreductases), or proteins [146]. At the intracellular level, metal ions interact with negatively charged components of the cell wall of microbes and then they are reduced into elemental atoms by enzymatic processes, assisted by coenzymes such as nicotinamide adenine dinucleotide (NADH)-dependent reductase [147]. In this regard, AgNPs prepared from bacteria such *Oscillatoria limnetica* have exhibited cytotoxic effects against colon cancer (i.e., HCT-116) and breast cancer (i.e., MCF-7) cell lines at 5.369 μg/mL and 6.147 μg/mL; in the same study, AgNPs inhibited the growth of drug resistant bacteria such as *Escherichia coli* and *Bacillus cereus* [148]. Similarly, AgNPs produced from fungi such as *Trametes ljubarskyi* and *Ganoderma enigmaticum* were cytotoxic against lung cancer (i.e., A549) and MCF-7 cell lines at 120 μg/mL [127].

### 4.2. Plant Extracts and Phytochemicals

On the other hand, plant extracts prepared from leaves, roots, stems, or flowers can reduce metal precursor salts into NPs through redox-mediated processes. This is attributed to the presence of amines, aldehydes, alcohols, and carboxylic acids in the structure of specific phytochemicals, such as terpenoids, flavonoids, polyphenols, and saponins [149]. For terpenoids such as eugenol, it has been reported that the deprotonation of its hydroxyl group leads to the appearance of an anion; its oxidation is driven by metal ions and is related to their reduction and formation into NPs [150]. Comparably, for flavonoids such as luteolin, it has been mentioned that their tautomeric transformation from the enol-form to the keto-form can release reactive hydrogen atoms and, thus, reduce metal ions into NPs [151,152]. 

For polyphenols such as gallic acid, their oxidation tendency leads to the initiation of the nucleation and growth stages of metallic NPs. In addition, the metal ion chelation capacity of other polyphenols such as caffeic acid and propyl gallate allows the formation of small AgNPs with applications against cancer [153]. Regarding applications against cancer, AgNPs prepared with the ethanol extract from *Artemisia turcomanica* exhibited apoptotic effects in gastric cancer cells (AGS) by increasing the expression of the pro-apoptotic protein Bax and decreasing the expression of the anti-apoptotic protein Bcl2 at 4.88 μg/mL [154]. In another study, AgNPs synthesized with stem extracts from medicinal plants such as *Commiphora gileadensis* were cytotoxic for colon cancer cell lines such as HCT116, HT29, and SW620 in a dose-dependent manner (10–100 μg/mL) [155]. In the same study, AgNPs increased the expression level of ATM, ATR, CHK1, and CHK2, which are genes related to DNA damage in cancer cells.

### 4.3. Amino Acids

Amino acids are organic compounds that act as building blocks for forming protein complexes. Structurally, they contain an amine group (-NH2), a carboxylic acid group (-COOH), and a sidechain [156]. In molecular and cellular processes, amino acids are necessary for protein synthesis, cell growth, and cell proliferation, and they can dictate complex events such as immune cell activation and functionality [157]. In nanomedicine, amino acids such as arginine, tyrosine, tryptophan, methionine, cysteine, and histidine, represent eco-friendly agents capable of reducing metal ions into stable NPs, by means of their functional groups, concentration, temperature of the system, and solution ionic strength [158,159]. For aromatic amino acids such as tyrosine and tryptophan, their phenol and indole groups are responsible for reducing silver ions into NPs. For arginine, it has been reported that its interaction with silver ions occurs through two main mechanisms. 

In the former, the carboxyl oxygen and nitrogen of the *α*-amino group can coordinate with silver ions and form complexes, also known as charge-solvated complexes [160]. In the latter, silver ions interact with the oxygen atoms of the guanidino group and form salt bridge complexes [160]. The formation of AgNPs with sulfur-containing amino acids arises from similar mechanisms. For instance, L-methionine can form complexes with silver ions, reduce them, and promote radical cation formation. The result can continue reacting with silver ions, lead to their reduction, and cause the formation of another complex [161]. In lung cancer treatment, the use of amino acids to synthesize AgNPs is limited; however, for other types of cancer, AgNPs synthesized with L-histidine can induce the apoptosis of SiHa cells (human cervical cancer cells) by inducing reactive oxygen species (ROS) and mitochondrial dysfunction [162].

## 5. Capping Agents

Owing to their functional groups, certain molecules can control the size of metallic NPs by preventing their overgrowth or aggregation during synthesis or storage. These compounds are known as capping agents, and they can also act as reducing agents and determine the bioactivity, cellular uptake, and physiochemistry of metallic NPs [163,164]. For AgNPs, capping agents can modify their behavior during characterization through spectroscopic methods such as UV-Vis and FTIR spectroscopy [165]. From a mechanistic point of view, capping agents act as barriers or coatings that, due to their cationic or anionic nature, can produce electrostatic interactions between particles during their formation and, thus, prevent their aggregation. This is also avoided due to the stearic hindrance caused by the length of polymeric or amino acidic capping agents [166,167]. Examples and functions of capping agents are represented in Figure 4.

There are various molecules that are currently used for this purpose; for example, amino acids (e.g., taurine) [168], surfactants (e.g., cetyl trimethylammonium bromide and sodium dodecyl sulfate), polymers (e.g., polyethylene glycol and polyvinylpyrrolidone) [169,170], organic acids (e.g., ethylenediaminetetraacetic acid, and oleic acid), proteins (e.g., bovine serum albumin, collagen, and gelatin), and natural compounds such as quercetin [166,171,172]. In therapeutics, the use of capping agents is necessary, since they can enhance the bioavailability, biocompatibility, and solubility of metallic NPs, with applications in cancer imaging and biosensing [166]. In addition, their implementation can protect the formed NPs against moisture, reduce their toxicity, and increase their affinity towards cell membranes (see Figure 4) [166,173]. 

In LC treatment, an aqueous extract from *Toxicodendron vernicifluum* was used as a reducing and capping agent for the synthesis of AgNPs. In that work, 320 μg/mL AgNPs induced the apoptosis of A549 cells by increasing ROS. In the same study, AgNPs inhibited the growth of *Helicobacter pylori* and Shiga toxin-producing *E. coli* (STEC) at 18.14 and 8.12 μg/mL, respectively [174].

In another study, embelin, a strong anticancer molecule distributed in *Embelia ribes*, was used as a reducing and capping agent for the synthesis of AgNPs. In the same report, embelin-AgNPs inhibited the proliferation of A549 cells in a dose-dependent manner (10–200 μg/mL) and caused their death through apoptosis [175]. Comparably, shikonin, a cytotoxic naphthoquinone originally isolated from the roots of *Lithospermum erythrorhizon*, was used to promote the formation of AgNPs. In this study, shikonin-AgNPs were labeled with radioactive iodine-131, to study their biodistribution in normal Swiss albino mice, which resulted in a high preferential uptake by lung tissues. In addition, they were cytotoxic against A549 cells at IC_50_ 2.4 μg/mL after 24 h [176]. For complex natural mixtures, the methanol extract from the leaves of *Juniperus polycarpos* was obtained to fabricate AgNPs that exhibited antiproliferative and antimetastatic properties against A549 cells. The capping effect of this extract on the AgNP surfaces was suggested by the FTIR results and changes in the ζ-potential value [177]. The activities of biosynthesized AgNPs against LC are presented in Table 3.

## 6. Possible Anti-Lung Cancer Mechanisms of Biosynthesized AgNPs

Anticancer agents can be classified according to their mode of action. For example, alkylating agents such as cyclophosphamide and ifosfamide cross-link with the N-7-guanine residues of DNA strands, leading to abnormal base pairing, inhibition of cell division, and cell death [192]. However, their use can cause numerous adverse effects, such as hemorrhagic cystitis, bone marrow depression, nausea, and vomiting [193]. 

On the other hand, anthracyclines such as doxorubicin, epirubicin, and idarubicin act as DNA intercalating agents that can inhibit DNA-dependent functions and induce the production of ROS [194]. In addition, they can arrest the cell cycle in G1/G2, induce apoptosis, and cause mitochondrial dysfunction when binding to topoisomerase 2 isoenzymes (i.e., Top2*α* and Top2*β*) [195]. Despite their widespread use in cancer treatment, the use of these compounds is mainly related to cardiotoxic effects and congestive heart failure [196]. For platinum-based compounds such as cisplatin, carboplatin, and oxaliplatin, it has been observed that they can cause lipid peroxidation and enzyme inhibition by disrupting calcium homeostasis. Moreover, they can induce cell cycle arrest and cell death by modulating the activity of serine/threonine kinases (AKT), mitogen activated protein kinases (MAPK), and c-Jun N-terminal kinases (JNKs) [197]. Even though some of these agents represent the first-line treatment for a variety of cancers, including LC, it is well-known that their administration can result in nephrotoxicity, myelosuppression, and neurotoxicity [198].

At the molecular level, AgNPs with different sizes can induce DNA fragmentation, protein carbonylation, and peroxidation of the lipid membrane of cancer cells by increasing the production of ROS [199]. In the same regard, biosynthesized AgNPs can alter the transmembrane potential of mitochondria by causing the overproduction of free radicals (i.e., hydrogen peroxide), which results in cell death by apoptosis [200]. At the cellular level, AgNPs can hinder the uncontrolled division or induce the death of cancer cells through autophagy or apoptosis via distinct signaling pathways, such as the nuclear factor-*κ*B (NF-*κ*B), COX-2, and PI3K/AKT/mTOR pathways [201,202]. In addition, treatment with AgNPs can inhibit cancer cells motility by means of their interaction with metalloproteinases (MMPs), arrest the cell cycle, cause multiple morphological changes (e.g., blebbing of cytoplasm and chromatin condensation), and modulate the expression of pro-apoptotic and anti-apoptotic proteins [203,204]. In tumor vessels, the anticancer activity of AgNPs is attributed to the enhanced permeability and retention (EPR) effect. In this process, AgNPs passively accumulate into the tumor interstitial space and disrupt the interaction between the cellular (e.g., cancer-associated fibroblast and macrophages) and non-cellular (e.g., the extracellular matrix) components of the tumor stroma. In consequence, metastatic, migrative, and proliferative activities can be decreased [205]. Some of these anticancer mechanisms are depicted in Figure 5.

The mechanisms described above have been reported in numerous studies. For instance, 13 nm AgNPs can induce the apoptosis of A549 cells in vitro by modulating the NF-*κ*B pathway and by inducing the overexpression of pro-apoptotic proteins (i.e., Bax and Bad), and decreasing the levels of anti-apoptotic proteins (i.e., Bcl-2) [206]. Comparably, 8–22 nm AgNPs synthesized with peel extracts from *Citrus maxima* inhibited the growth and caused the death of H1299 cells by apoptosis by decreasing the activity of the NF-*κ*B pathway and increasing the levels of caspase-3 and survivin [207]. In the same, study, treatment with AgNPs suppressed the growth of H1299 tumors in severe combined immunodeficient (SCID) mice [207]. 

In another study, AgNPs (~10–30 nm) synthesized with powders from *Salvia* species (i.e., *Salvia coccinea*, *Salvia leucantha*, and *Salvia splendens*) caused the death of A549 cells by apoptosis at 402, 364, and 418 μg/mL, respectively [185]. In the same cell line, AgNPs caused multiple morphological changes such as cytoplasmic blebbing, chromatin fragmentation, and nuclear swelling [185]. On the other hand, AgNPs (3–10 nm) prepared with ethanol extracts from conventional herbal products such as garlic (*Allium sativum* L.), inhibited the proliferation of A549 cells in a dose-dependent manner; this event was attributed to the induction of ROS and cellular damage [208]. Similarly, 18–39 nm AgNPs synthesized with the supernatant of *Streptomyces hirsutus* strain SNPGA-8 were cytotoxic at IC_50_ 31.4 μg/mL against A549 cells. In the same study, treatment with AgNPs increased ROS production in the same cell line and exhibited antibacterial properties against pathogenic Gram-positive (e.g., *Staphylococcus aureus*), Gram-negative (e.g., *Pseudomonas aeruginosa*), yeast (e.g., *Candida glabrata*), and fungal (e.g., *Fusarium oxysporum*) strains [209].

In tumor growth, angiogenesis is a key process that supports the supply of oxygen and nutrients. There are numerous proteins and receptors that drive this process, for example, vascular endothelial growth factor receptor (VEGFR), platelet-derived growth factor (PDGFR), and fibroblast growth factor receptor (FGFR) [210]. In SCLC treatment, monoclonal antibodies such as anlotinib, sunitinib, nintedanib have been tested as inhibitors of VEGFR, PDGFR, and FGFR, respectively. However, during their evaluation, refractory LC, anorexia, hypertension, and pulmonary hemorrhages were registered [211]. In contrast, angiogenesis inhibitors (e.g., vandetanib, sorafenib, nintedanib, and ramucirumab) that also target some of these receptors in NSCLC have been tested in combination with chemotherapeutic agents such as carboplatin, paclitaxel, and docetaxel. However, adverse events such as anorexia, diarrhea, fatigue, and hypertension have been also documented [212]. As an alternative, multiple models have been developed to validate the anticancer activities of AgNPs; for example, lung monocultures, lung co-cultures, lung 3D cultures, and ex vivo and in vivo models [213].

The chorioallantoic membrane (CAM) assay is an in vivo model used to evaluate the anti-angiogenic properties, biocompatibility, and toxicity of organic or inorganic NPs [214,215]. In addition, the capacity of metallic NPs (i.e., gold and silver) to stimulate or inhibit angiogenic processes can be studied using ex vivo models such as the rat aortic ring model [216,217]. In the biological synthesis of AgNPs against LC, the use of these models is limited; however, it has been helpful to expand the knowledge about their therapeutic performance. For example, 100 nm AgNPs synthesized with an aqueous extract of the brown alga *Dictyota ciliolata* were reported to have antiangiogenic activity by inhibiting the formation of tertiary blood vessels in a CAM assay using A549 cells. In the same cell line, the produced AgNPs induced DNA fragmentation, inhibited cell migration, and where cytotoxic at IC_50_ 5 μg/mL [218]. On the other hand, 16.5 nm AgNPs synthesized with *Saliva officinalis* aqueous extract compromised the vascular organization of the CAM model by inhibiting blood vessel formation and exhibiting cytotoxic effects on endothelial cells in a dose-dependent manner (50–150 μg/mL) [219].

## 7. Discussion 

Among the top-five types of cancer worldwide, LC has remained the most problematic since 1985, in respect to its incidence, diagnosis, and treatment [220]. The incidence of LC has been registered in multiple countries distributed in the Middle East, Eastern Europe, and Southeast Asia [221]. The regions with the highest incidence among men are Eastern Asia, and Central and Eastern Europe [25]. For women, the highest incidence of LC has been observed in North America, Northern Europe, and Western Europe [25]. The discrepancies between the incidence of LC between men and women can be explained by differences in smoking patterns and presence of gene polymorphisms. 

For instance, women possess a higher risk to develop AC since they are more likely to develop mutations in genes (e.g., *CYP1A1*, *EGFR*, and *KRAS*) related to the formation of DNA adducts and carcinogenesis [222]. In contrast, men with a history of smoking are more likely to display mutations in the *MET*, *PIK3CA*, *KRAS*, and *CDKN2A* genes, which can contribute to the manifestation of SqCC [223,224]. The relationship between high consumption of cigarettes and SqCC development can be attributed to the fact that high exposure to polycyclic aromatic hydrocarbons and tobacco-specific N-nitrosamines promotes carcinogenic processes [225].

Current approaches to the diagnosis and treatment of LC rely on the clinical status of the patient, size and location of the primary tumor, presence or not of metastasis, and the type of LC [226]. As in other types of cancer, the treatment of LC is based on extensive chemotherapy, radiotherapy, immunotherapy, and targeted therapy regimens. However, their administration can cause severe toxic effects among patients, discomfort, drug resistance, and development of secondary tumors. In this sense, the administration of taxanes such as paclitaxel and docetaxel, which are common cytotoxic agents administered in patients with advance NSCLC, has been corelated to bilateral pulmonary interstitial infiltrates, cough, fever, pneumonitis, and even fatal cases [227]. These adverse effects are manifested due to the poor solubility and lack of tumor specificity of both taxanes [228]. Comparably, first-line chemotherapeutics such as cisplatin and carboplatin have been related to myelosuppression, nephrotoxicity, healthy tissue damage, and neurotoxicity [229]. The adverse effects caused by platinum-based compounds depend upon the dose, individual factors (e.g., age, and diet), and resistance mechanisms [230].

Drug resistance mechanisms contribute to disease progression and are developed intrinsically or externally by cancer cells to avoid the activity of anticancer drugs and for survival. The resistance of cancer cells against the current treatment modalities arises from alterations in the tumor microenvironment, tumor hypoxia, genetic mutations, epigenetic changes, and aberrations in molecular and cellular events [231]. In LC cells, the presence of such mechanisms can vary in respect to the subtype of LC and the time of clinical presentation. Despite of the tumor heterogeneity among SCLC and NSCLC tumors, drug resistance mechanisms have developed, reducing the efficacy of the current chemotherapeutics (e.g., etoposide, cisplatin, and carboplatin), radiotherapy (e.g., PCI) modalities, immunotherapeutics (e.g., atezolizumab and nivolumab), and targeted therapy (e.g., *ALK* and *EGFR* inhibitors) agents [229]. 

Since treatment response among LC patients is compromised by genetic, molecular, and pharmacological factors, new approaches are needed, to provide safe and precise modern therapeutic modalities against this disease. The term nanotechnology was introduced in 1959 by Dr. Richard Feynman, and later it was defined by Dr. Norio Taniguchi as the processing and separation, consolidation, and deformation of materials by one atom or one molecule in 1974 [232]. The use of nanotechnology in medicine is referred to as nanomedicine and has been exploited in multiple fields such as cardiology, dermatology, genetics, and oncology, to provide early diagnoses and high specificity treatment regimens against a wide array of diseases [233,234]. For biological applications, nanomedicine needs to rigorously characterize the physiochemical features (e.g., surface area, porosity, and crystal structure), physiological behavior (e.g., dissolution rate, kinetics, and water solubility), and quality of NMs [235]. In comparison to other nanomedicines, NPs have been exploited due to their high surface area to volume ratio and spherical shape, they can exhibit improved aqueous solubility, higher half-life for clearance, specificity for cellular receptors, and an improved targeting and loading capacity [236].

The anticancer properties of metallic NPs can vary in accordance with various factors, such as their size, shape, ζ-potential, and synthesis condition. It has been documented that small (<100 nm) NPs are considered suitable anticancer agents, as their size enables effective delivery, EPR effect, biocompatibility, and stability [237,238]. On the contrary, large (>200 nm) NPs are predisposed to agglomerate in the liver and remain for longer periods in blood [239]. In Table 1, we showed applications of metal-based NPs where, alone or in combination with drugs or vitamins, they can exhibit numerous anticancer activities against breast, hepatocellular, human neuroblastoma, and esophageal cancer cell lines; whereas, in Table 2, we sought to summarize findings about the properties of biosynthesized AgNPs against cancer models. The variations in the therapeutic performance of the reviewed metal-based NPs could be due to the intrinsic features of the noble metal, surface functionalization capacity, size, and shape [240].

The access to current first line therapy regimens is problematic, due to their high cost and limited access among middle- and low-income countries [241]. In recent years, it has been estimated that the cost of anticancer drugs has increased from USD 30,447 to USD 161,141 [242], and this is expected to increase even more due to the introduction of modern therapies such as chimeric antigen receptor-engineered T cell (CAR-T cell) therapy [243,244]. In the same context, the affordability of cancer therapy can be challenging, since its cost can depend upon the number of doses, therapy duration, treatment modality, type of cancer, geographic region, and appearance of new classes of drugs that do not necessarily improve the clinical outcome of patients [245,246]. Thus, therapeutic alternatives able to meet the demand for affordability and overcome the high prices of anticancer drugs are needed.

In contrast to the high costs of anticancer drugs, biosynthesized NPs represent a cost-effective alternative, since the reagents used for their production are affordable and their formation can be achieved using unsophisticated laboratory materials [247]. For instance, AgNPs are preferably produced using reasonable amounts of reducing or capping agents that are widely available, solvent mediums that are non-toxic and accessible (i.e., water), and temperatures that do not necessarily have to be high and are achievable in a laboratory [248]. Moreover, the biosynthesis of AgNPs is also an effective and economical alternative for fabricating nanomedicines, since they can be considered for cancer treatment but also for other medical and non-medical sectors, such as antibacterial, antifungal, and anti-inflammatory agents [249]. However, the cost of their production could be elevated if costly materials, additional synthesis steps such as manipulation of metabolism microbial cell cultures [250], and characterization and purification techniques are applied [251]. 

The large-scale production of nanostructures with biomedical implications is a process necessary for their commercialization and clinical evaluation. In this case, parameters such as the drug loading capacity, polydispersity, size, shape, bioactivity, and possible toxicity of NPs need to be defined [252]. For the industrialization of biosynthesized AgNPs with applications in cancer treatment and diagnosis, the broad availability and accessibility of biological sources such as microbial cell cultures, plants, or natural products, enables their easy scale-up [253]. However, to avoid variabilities in their production at high yields, promote their reproducibility, and develop a cost-effective process, it would be convenient to establish a detailed program where their production in a sterile form, the necessary reagents, storage conditions, administration, and potential biological and environmental hazards are assessed [254,255]. 

Against LC cells, we reviewed several reports, where 10–50 nm AgNPs inhibited their proliferation and induced their death by increasing the generation of ROS, upregulating signaling pathways (e.g., mitochondrial pathway), enhancing enzyme activity (e.g., caspase 3 and 9), inducing morphological changes to cellular components (e.g., chromatin condensation), or inhibiting key carcinogenic phenomena. In addition, as presented in Table 3, biosynthesized AgNPs also compromise the viability of LC cell lines by modulating signaling pathways such as the EGFR/p38, which is commonly related to cancer tumorigenesis processes. In the same sense, biosynthesized AgNPs can lead to LC cell death by disrupting the structural integrity of mitochondria, which can be attributed to their capacity for enhancing the production of ROS. Regarding the use of biosynthesized AgNPs against LC in vivo models, the evidence is limited, since to the best of our knowledge, they have been only evaluated using CAM assays. Therefore, we consider this a research field that must be developed, in order to continue assessing the therapeutic potential of biosynthesized AgNPs.

The ζ-potential is a crucial parameter that, according to its magnitude (+30 or −30 mV), can confer AgNPs stability and resistance against aggregation. In nanomedicine, it is known that ζ-potential also influences the cellular uptake and capacity of NPs to induce cellular damage, interact with cellular components (e.g., organelle membranes and cell surface), and exhibit their bioactivities [256]. In Table 2, we presented various studies where negatively-charged biosynthesized AgNPs elicited distinct anticancer effects against multiple cancer cell lines. Moreover, we compiled the recent studies where biosynthesized AgNPs were cytotoxic against LC models, but also inhibited the growth of pathogenic bacteria at low concentrations. The dual antibacterial effect of biosynthesized AgNPs can be attributed to their negative ζ-potential, which can influence bacterial damage and cell death [257,258]. In addition, we consider that the capacity of biosynthesized AgNPs to act against microbial pathogens represents another promising feature in LC treatment, since patients are prone to develop infections caused by Gram-negative bacteria such as *E. coli*, *S. aureus*, and *P. aeruginosa* [259]. 

As part of green chemistry, biological sources such as amino acids, microbial cell cultures (e.g., bacteria, fungi, and viruses), plant extracts, phytochemicals, and enzymes are utilized to synthesize and stabilize anticancer AgNPs. The use of each category possesses various advantages and disadvantages that must be considered. For instance, plant extracts as reducing agents represent a low-cost and sustainable alternative to synthesizing bioactive AgNPs; however, additional approaches are needed to determine the exact compounds responsible for the reduction of AgNO_3_ ions into NPs [260,261]. Similarly, the use of microbial cell cultures (e.g., *Saccharomyces cerevisiae*) can be a cost-effective and nontoxic option for accelerating production at high scale and for conjugating functional groups onto the surface of AgNPs [262]. However, their implementation is also associated with important drawbacks; for example, it is a time-consuming process that can contaminate and alter the size, shape, and stability of the formed NPs [263].

Despite its innovating applications in nanomedicine, the biosynthesis of noble metal-based NPs (e.g., gold, palladium, and platinum NPs) can be a demanding process, in terms of material availability, product quality control, and synthesis conditions. It has been documented that the formation of metal-based NPs through biosynthesis can be a time- and energy-consuming process, where variabilities in the amount of reducing agents (i.e., plant extracts) can result in aberrant NPs with undefined sizes and shapes [264]. In this regard, the therapeutic performance, toxicity, and adequate formation of NPs biosynthesized with microbial cell cultures can depend upon the microorganism strain, pH, growth medium, and growth stage of the cultured cells [265]. The scaling-up and consequent translation into the pharmaceutical industry of biosynthesized NPs also has challenges, since the optimal synthesis conditions, yield, stability, pharmacodynamics, and pharmacokinetics of biosynthesized metal-based NPs have been poorly studied [266]. 

## 8. Conclusions 

In conclusion, biosynthesized AgNPs are promising therapeutic agents that have been tested against various types of cancer. Against LC models, biosynthesized AgNPs have been predominantly studied through cytotoxic assays; however, such results can be considered as a relevant scientific precedent to look for new research opportunities, where the anti-lung cancer properties of biosynthesized AgNPs can be assessed.

AgNPs synthesized from biological sources can interfere with their cell cycle, modulate signaling pathways, induce cell death, and enhance the production of intracellular molecules. Although the literature about the activity of biosynthesized AgNPs in in vivo models is scarce, it has been demonstrated that they can interfere with tumor development and major carcinogenic processes at low concentrations. However, further approaches are needed to test the efficacy of biosynthesized AgNPs against other LC models. Due to their cost-effectiveness and low toxicity, it seems that the common biological sources for producing AgNPs include plant extracts and microbial strains. There is a need to test the capacity of other molecules to induce the formation of these therapeutic nanostructures. As a matter of fact, the search for novel methods to fabricate noble metal-based nanomaterials with therapeutic applications has been an active research field over the last decade. However, the design of innovative roadmaps to appraise the major pharmacodynamic, pharmacokinetic, and toxicological parameters of biosynthesized AgNPs is urgently required for future drug development and translation into clinical trials. In the foreseeable future, the combination of biosynthesized AgNPs with other nanostructures could be exploited to continue expanding the knowledge about nanometric materials in the medical field. For the industrial translation of biosynthesized AgNPs, synthesis conditions must be optimized, defined, and improved for further applications at large scale. In the same regard, there is an urgent need to design economic studies about the affordability and costs of the production of biosynthesized AgNPs.

## Figures and Tables

**Figure 1 biomedicines-11-00389-f001:**
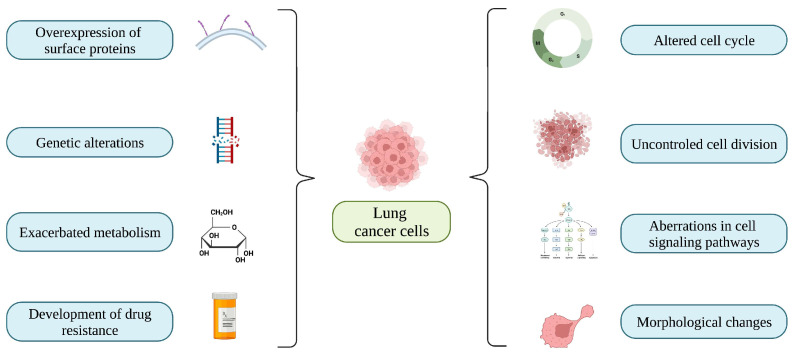
Features of lung cancer cells: overexpression of surface proteins, genetic alterations, exacerbated metabolism, drug resistance, altered cell cycle, uncontrolled cell division, aberrations in cell signaling pathways, and morphological changes.

**Figure 2 biomedicines-11-00389-f002:**
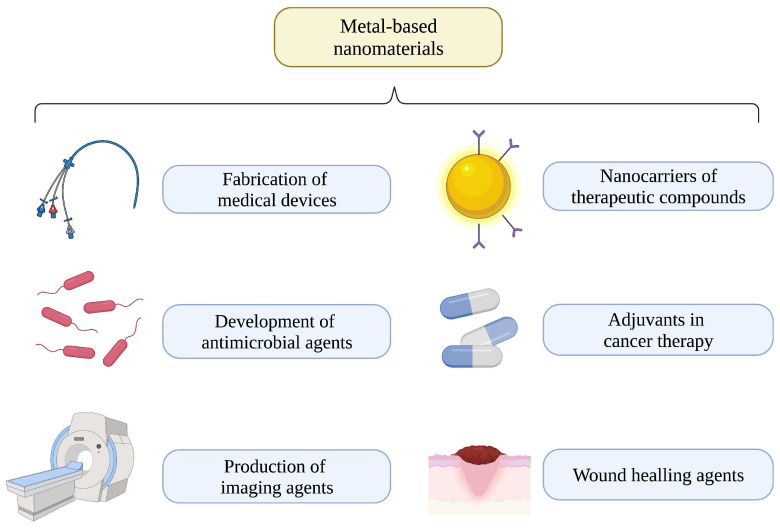
Applications of metal-based nanomaterials.

**Figure 3 biomedicines-11-00389-f003:**
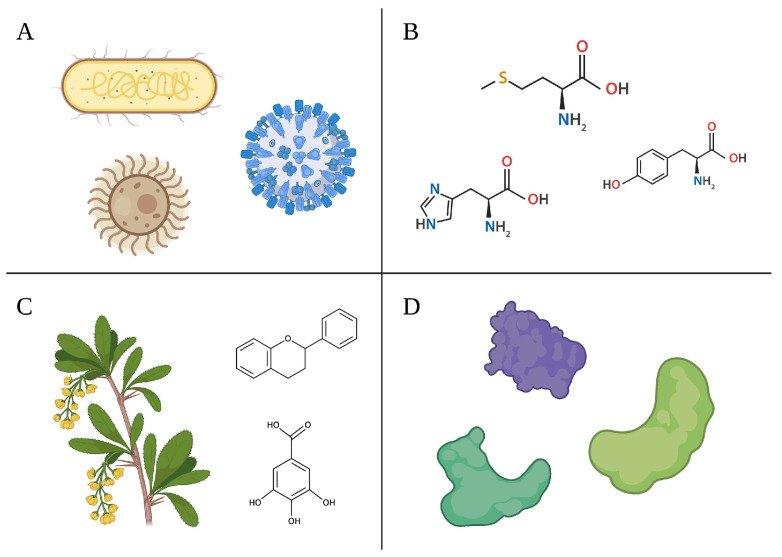
Common biological sources for synthesizing AgNPs: (**A**) microorganisms, (**B**) amino. acids, (**C**) plant extracts and phytochemicals, and (**D**) proteins.

**Figure 4 biomedicines-11-00389-f004:**
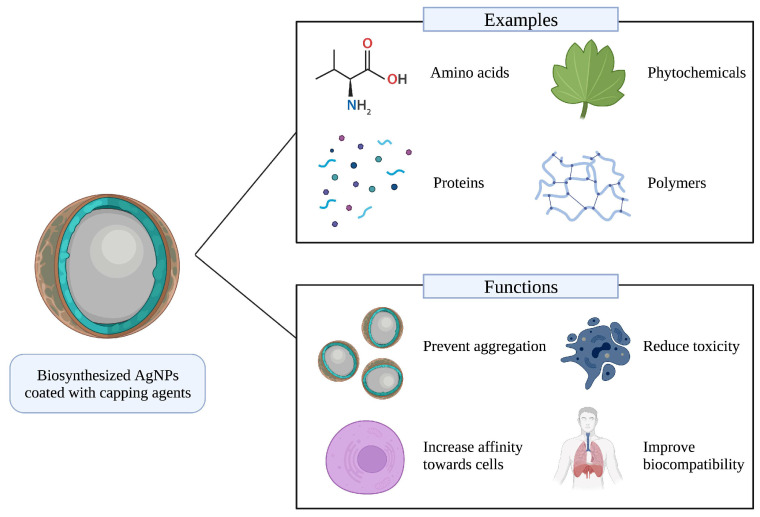
Examples and functions of capping agents in biosynthesized AgNPs.

**Figure 5 biomedicines-11-00389-f005:**
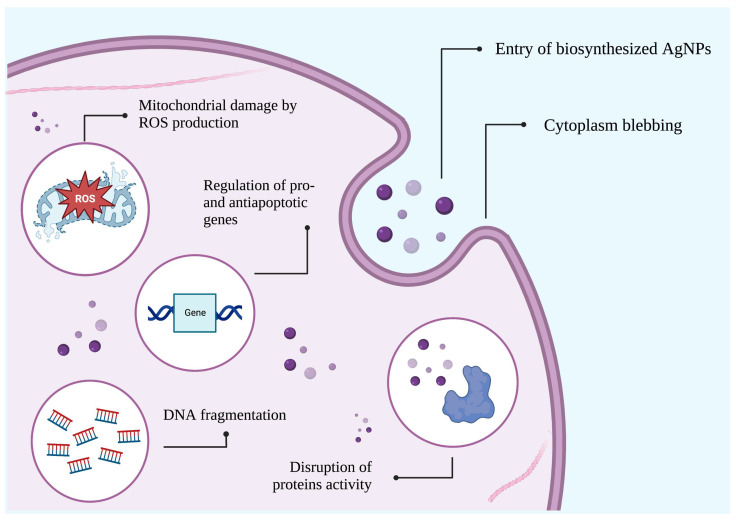
Possible anticancer mechanisms of biosynthesized AgNPs.

**Table 1 biomedicines-11-00389-t001:** Applications of noble metal-based nanoparticles against different types of cancer.

Noble Metal	Precursor Salt	Size (nm)	Shape	Results	Reference
Silver	AgNO_3_	13	Spherical	AgNPs exhibited cytotoxic effects against breast and hepatocellular carcinoma cancer cell lines by increasing the production of ROS, upregulating the expression of pro-apoptotic (caspase 3, Bax, and p53), and downregulating the expression of anti-apoptotic genes (Bcl2).	[91]
Silver	AgNO_3_	31	Spherical	AgNPs synthesized with *Citrullus colocynthis* were cytotoxic against human epidermoid larynx carcinoma cell lines. Treatment with AgNPs caused apoptosis by elevating caspase 3 levels and caused DNA fragmentation.	[92]
Silver	AgNO_3_	10	Spherical	AgNPs were synthesized using resveratrol and combined with gemcitabine to treat human ovarian cancer cells. Treatment with AgNPs and gemcitabine inhibited the proliferation and viability of human ovarian cancer cells. The combination of AgNPs with gemcitabine also enhanced the apoptotic activity.	[93]
Silver	AgNO_3_	22	Spherical	AgNPs were cytotoxic against breast cancer cell lines. Treatment with AgNPs compromised cell membrane integrity and induced oxidative stress and apoptosis.	[94]
Gold	HAuCl_4_	27	Spherical	Varlitinib was delivered into pancreatic cancer cell lines by AuNPs functionalized with PEG. At low concentrations (IC_50_ 80 nM), PEG-AuNPs increased the toxic effect of varlitinib and exhibited cytostatic effects against pancreatic cancer cell lines.	[95]
Gold	HAuCl_4_·3H_2_O	40	Oval and spherical	Water extracted from *Rhus chinensis* was used to biosynthesize AuNPs. AuNPs were cytotoxic against adenocarcinoma, hepatocellular carcinoma, and osteosarcoma cancer cell lines in a dose-dependent manner.	[96]
Gold	HAuCl_4_	200	Hexagonal and triangular	AuNPs were biosynthesized using *Eclipta* extract. AuNPs were used to construct a drug delivery system containing doxorubicin. AuNPs nanoconstruct with doxorubicin was cytotoxic against human breast cancer cell lines.	[97]
Copper	Cu(NO_3_)_2_·3H_2_O	13	Spherical	CuNPs were cytotoxic against various esophageal cancer cell lines (i.e., ESO26, OE33, and KYSE-270) in a dose-dependent manner (1–1000 μg/mL). CuNPs also exhibited antioxidant effects against DPPH free radicals at 300 and 105 μg/mL.	[98]
Copper	CuSO_4_·5H_2_O	42	Spherical	CuNPs were synthesized utilizing *Manilkara zapota* leaf extract. Treatment with CuNPs was cytotoxic against human breast cancer cells and exhibited antimicrobial activity against bacterial and fungal plant pathogens.	[99]
Copper	CuSO_4_·5H_2_O	18	Hemispherical	CuNPs were biosynthesized using aqueous *Tilia* extract. CuNPs were cytotoxic against human breast cancer, human hepatic cancer, and human colon cancer cell lines. CuNPs inhibited the growth of pathogenic Gram-positive and Gram-negative bacteria.	[100]
Copper	Cu(NO_3_)_2_·5H_2_O	63	Spherical	CuNPs were used to produce a nanocomposite based on albumin. CuNP nanocomposite was highly cytotoxic through enhancing the production of ROS and inducing apoptosis in triple-negative breast cancer cell lines.	[101]
Palladium	PdCl_2_	5	Spherical	PdNPs induced cancer cell death of human cancer cells by inducing autophagy, increasing ROS production, promoting caspase 3 activity, enhancing the leakage of LDH, and impairing mitochondrial membrane potential in a dose-dependent manner (0–10 μg/mL).	[102]
Palladium	PdCl_2_	7	Spherical	PdNPs were biosynthesized using *Urtica* extract. PdNPs were cytotoxic against human colon cancer, human breast cancer, and human pancreatic cancer cells lines. PdNPs inhibited the growth of pathogenic Gram-positive and Gram-negative bacteria.	[103]
Palladium	PdCl_2_	25	Spherical	PdNPs were biosynthesized using R-phycoerythrin and combined with tubastatin A in a synergistic approach. In triple-negative breast cancer cells, the combination of PdNPs with tubastatin A increased caspase 3 activity, caused DNA fragmentation, elevated the expression of pro-apoptotic genes, reduced cell viability, and decreased mitochondria transmembrane potential.	[104]
Palladium	PdCl_2_	10	Spherical	Human lung cancer cells were treated with PdNPs combined with melatonin. Treatment with PdNPs combined with melatonin decreased mitochondrial transmembrane potential, induced apoptosis-caused DNA damage, and elevated the expression levels of apoptotic genes.	[105]
Platinum	PtCl_6_	25	Spherical	PtNPs were combined with RA as an alternative to combination therapy against neuroblastoma. The treatment of PtNPs in combination with RA induced apoptosis-associated networks, DNA damage, mitochondrial dysfunction, OS, and ERS in human neuroblastoma cells. PtNPs also exhibit other shapes, such as cubic, oval, triangle, and rod-like.	[106]
Platinum	K_2_PtCl_6_	45	Spherical	*Streptomyces* sp. Culture was used to biosynthesize PtNPs. PtNPs decreased the cell viability of human breast cancer cell lines in a dose-dependent manner.	[107]
Platinum	K_2_PtCl_6_	34	Spherical	PtNPs were cytotoxic against human breast cancer and human hepatoma cancer cell lines in a dose-dependent manner.	[108]
Platinum	H_2_PtCl_6_·6H_2_O	113	Spherical	PtNPs were biosynthesized using the leaf extract of *Psidium guajava*. Treatment with PtNPs inhibited the proliferation of human breast cancer cell lines by arresting their cell cycle at the G0/G1 phase. Treatment with PtNPs also inhibited the migration and decreased the cell viability of human breast cancer cell lines. Treatment with PtNPs inhibited the growth of pathogenic Gram-negative bacteria.	[109]

Abbreviations: AgNO_3_, silver nitrate; HAuCl_4_, chloroauric acid; HAuCl_4_.3H_2_O, gold (III) chloride trihydryte; PEG, polyethylene glycol; IC_50_, half maximal inhibitory concentration, PdCl_2_, palladium(II) chloride; LDH, lactate dehydrogenase; Cu(NO_3_)_2_, copper nitrate trihydrate; CuSO_4_·5H_2_O, copper sulfate pentahydrate, Cu(NO_3_)_2_·5H_2_O, copper nitrate pentahydrate; DPPH, 2,2-diphenyl-1-picrylhydrazyl; ROS, reactive oxygen species; HIF-1*α*, hypoxia-inducible factor 1 subunit alpha; PtCl_6_, Hexachloroplatinate; K_2_PtCl_6_, dipotassium hexachloroplatinate H_2_PtCl_6_·6H_2_O, chloroplatinic acid hexahydrate; RA, retinoic acid; OS, oxidative stress; ERS, endoplasmatic reticulum stress.

**Table 2 biomedicines-11-00389-t002:** Biosynthesized AgNPs with applications against cancer models.

Reducing Agent	Size (nm)	Shape	ζ-Potential (mV)	Results	Reference
*Ginkgo biloba* leaves extract	40	Spherical or oval	−34.5	Treatment with AgNPs inhibited the growth and proliferation of cervical cancer cells. Treatment with AgNPs decreased cellular attachment and increased cellular atrophy. Treatment with AgNPs induced apoptosis via the mitochondrial pathway and generation of intracellular ROS.	[132]
*Teucrium polium* extract	100	Spherical	N.I.	Treatment with AgNPs was cytotoxic against human gastric cancer cell lines in a dose-dependent manner. Several possible mechanisms of action have been suggested: elevation of ROS, protein damage, oxidative stress.	[133]
*Artemisia vulgaris* extract	25	Spherical	N.I.	Treatment with AgNPs was cytotoxic against human cervical cancer and human breast cancer cell lines. AgNPs inhibited the froth of pathogenic Gram-positive and Gram-negative bacteria. AgNPs exhibited DPPH and superoxide radical scavenging activity in a dose-dependent manner.	[134]
*Alternanthera sessilis* extract	50	Spherical	N.I.	AgNPs were cytotoxic against human prostate cancer and human breast cancer cell lines in a time- and dose-dependent manner. AgNPs induced the apoptosis of the tested cell lines by causing cell shrinkage, coiling, and oxidative stress.	[135]
*Cucumis prophetarum* aqueous leaf extract	90	Spherical	−36.7	AgNPs were cytotoxic against human triple-negative breast cancer, human hepatoma cancer, and human lung cancer cell lines. AgNPs inhibited the growth of Gram-positive and Gram-negative bacteria. AgNPs also showed higher free radical inhibition capacity in DPPH and ABTS assays.	[136]
*Dimocarpus Longan Lour.,* peel extract	32	Cubic	N.I.	AgNPs were cytotoxic against human prostate cancer cells by elevating the activity of caspase 3 and decreasing the expression of Bcl 2, survivin, and STAT 3. AgNPs exhibited bactericidal properties against Gram-positive and Gram-negative bacteria strains.	[137]
*Taxus yunnanensis* extract	27	Crystal	−28	AgNPs were cytotoxic and induced the apoptosis of human hepatoma cancer cells. AgNPs inhibited the growth of pathogenic Gram-negative bacteria. The ζ-potential of AgNPs was evaluated after two weeks.	[138]
*Cyperus conglomeratus* root extract	468	Spherical	N.I.	AgNPs were cytotoxic against human breast cancer cell lines but not against normal fibroblast cells. AgNPs induced the apoptosis of human breast cancer cell lines by upregulating the levels of Bax and downregulating the expression of YAP and Bcl 2.	[139]
*Lyngbya majuscula* strain	149	Spherical	−35.2	AgNPs diminished the cell viability of human leukemic cell lines in a dose- and time-dependent manner. AgNPs were not cytotoxic against peripheral blood mononuclear cells. Treatment with AgNPs was also effective against *P. aeruginosa*.	[140]
Walnut green husk aqueous extract	31	Spherical	−33.8	AgNPs were cytotoxic against breast cancer and fibroblast cell lines. AgNPs exhibited antibacterial activity against standard and nosocomial strains. AgNPs possessed a high radical scavenging ability in a time- and dose-dependent manner.	[141]
*Talaromyces purpurogenus* extracellular pigment	41		−24.8	AgNPs were cytotoxic against human cervical cancer, human liver cancer, and human embryonic kidney cell lines. Among human liver cancer cell lines, treatment with AgNPs disrupted cell membrane integrity and compromised their morphology in a dose-dependent manner. AgNPs inhibited the growth of *S. epidermis* and *E. coli*.	[142]
Actinobacterial strain SH11	16	Spherical	−17.1	Treatment with AgNPs was cytotoxic against breast cancer and macrophage cells in a dose-dependent manner. Treatment with AgNPs induce the production of ROS in the tested cell lines.	[143]
*Nocardiopsis* sp. MBRC-1	45	Spherical	N.I.	The supernatant of *Nocardiopsis* sp. MBRC-1 was used to biosynthesize AgNPs. Treatment with AgNPs was cytotoxic against human cervical cancer cell lines in a dose-dependent manner. Gram-positive and Gram-negative bacteria, as well as fungi strains, were susceptible to AgNP treatment.	[144]
*Penicillium shearii* AJP05	8.0	Spherical	N.I.	AgNPs induced cytotoxicity of human osteosarcoma and human hepatocellular carcinoma cells by inducing ROS generation and apoptosis. AgNPs sensitized resistant cells to cisplatin and increased the caspase 3 enzyme activity in a dose-dependent manner. AgNPs enhanced cell death by inhibiting autophagy and inducing ROS production.	[145]

Abbreviations: AgNPs, silver nanoparticles; ROS, reactive oxygen species; nm, nanometers; mV, millivolts; DPPH, 2,2-diphenyl-1-picrylhydrazyl; ABBT, 2,2′-azino-bis(3-ethylbenzothiazoline-6-sulfonic acid; YAP, yes-associated protein; N.I., not indicated.

**Table 3 biomedicines-11-00389-t003:** Activities of biosynthesized AgNPs against LC.

Noble Metal	Precursor Salt	Reducing Agent	Shape	Activities	Reference
Silver	AgNO_3_	*Populus nigra* L. extract	Spherical, rhombohedral, and triangular	AgNPs diminished the proliferation human lung carcinoma epithelial cells in a dose-dependent manner.	[178]
Silver	AgNO_3_	*Phoenix dactylifera* extract	Spherical	Treatment with AgNPs arrested the cell cycle in the sub-G1 phase, induced the generation of ROS, and caused the loss of mitochondrial membrane potential of human lung adenocarcinoma cells.	[179]
Silver	AgNO_3_	*Botryodiplodia theobromae* mycelium	N.I.	Treatment with AgNPs reduced the proliferation of human lung cancer cells lines in a cell viability assays.	[180]
Silver	AgNO_3_	*Azadirachta indica* extract	Spherical	AgNPs increased the generation of ROS and reduced the viability of NSCLC cancer cell lines. Treatment with AgNPs was less toxic against brine shrimp nauplii and human red blood cells.	[181]
Silver	AgNO_3_	*Cleistanthus collinus* extract	Spherical	Treatment with AgNPs inhibited the viability of human lung cancer cells in a dose-dependent manner.	[182]
Silver	AgNO_3_	*Thelypteris glandulosolanosa* extract	Spherical	Treatment with AgNPs reduced the viability of lung epithelial carcinoma cell lines.	[183]
Silver	AgNO_3_	*Origanum vulgare*	Spherical	The formed AgNPs reduced the cell viability of human lung cancer cell lines in a dose-dependent manner.	[184]
Silver	AgNO_3_	*Salvia coccinea, Salvia leucantha, Salvia splendens* extracts	Spherical	Treatment with AgNPs decreased the viability of lung cancer cells. Among lung cancer cell lines, AgNPs induced apoptosis and caused several morphological changes: nuclear swelling, cytoplasmic blebbing, and chromatin fragmentation.	[185]
Silver	AgNO_3_	*Cratoxylum formosum Mucuna birdwoodiana* *Lindera strychnifolia* extracts	Spherical	Treatment with formed AgNPs compromised the viability of lung cancer cell lines by inducing their early and late apoptosis. AgNPs also exhibited wound healing and antioxidant activities.	[186]
Silver	AgNO_3_	*Juniperus chinensis* extract	Spherical	Treatment with AgNPs reduced the proliferation and induced the apoptosis of human lung cancer cells by upregulating the activities of caspase 3 and 9, as well as p53. The activity of AgNPs against human lung cancer cell lines was also attributed to the elevation of ROS.	[177]
Silver	AgNO_3_	*Gloriosa superba* L. extract	Spherical	AgNPs induced the apoptosis and decreased the proliferation of human lung cancer cell lines.	[187]
Silver	AgNO_3_	*Caulerpa taxifolia* extract	Spherical	Treatment with AgNPs reduced the viability and altered the morphology of human lung cancer cells.	[188]
Silver	AgNO_3_	*Toxicodendron vernicifluum* extract	Spherical and oval	Treatment of AgNPs decreased the viability of human lung cancer cell lines by inducing apoptosis and elevating the production of ROS.	[174]
Silver	AgNO_3_	*Dendropanax morbifera* Léveille extract	N.I.	Treatment with AgNPs reduced the proliferation of human lung cancer cells by inducing apoptosis and enhancing the production of ROS. AgNPs also reduced the migration of the tested cell lines, modified the EGFR/p38 pathway, and induced morphological changes.	[189]
Silver	AgNO_3_	*Avicennia marina* extract	Spherical	Treatment with AgNPs against human cancer cell lines induced the production of ROS, caused morphologic changes, and induced apoptosis by damaging the mitochondrial membrane.	[190]
Silver	AgNO_3_	*Tabebuiaroseo-alba* extract	Spherical	AgNPs lead human cancer cell lines to apoptosis by causing oxidative stress. Treatment with AgNPs caused DNA fragmentation and activated caspase 3 and 9.	[191]

Abbreviations: AgNPs, silver nanoparticles; ROS, reactive oxygen species; nm, nanometers; NSCLC, non-small cell lung cancer; EGFR, epidermal growth factor receptor; N.I., not indicated.

## Data Availability

Not applicable.

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
