# Peer review of "Activities against Lung Cancer of Biosynthesized Silver Nanoparticles: A Review"

_biomedicines, 2023, doi:10.3390/biomedicines11020389_

Round 1
Reviewer 1 Report
Jorge L. Mejía-Méndez et al. summarized the preparation and application of biosynthesized silver nanoparticles for lung cancer therapy. The manuscript was well written and fell within the scope of Biomedicines. However, there were some flaws deterring the acceptance of current version, particularly in the logic. A Major Revision should be conducted before a final decision. The detailed comments were as follows.
(1) In the Introduction Section, the reason why AgNPs were picked up for this review should be stated. In other words, what were the advantages of AgNPs over other NPs?
(2) As stated by the authors, Google Scholar, PubMed, and Springer databases were consulted in this work. This was somewhat similar to a bibliometrics’ methodology. Maybe the authors could provided some bibliometric data about the retrieved literature?
(3) Please reconsider the significance of the historical review about noble metal NPs (Section 3). As the authors had focused on AgNPs according to the Title, Abstract and Introduction, it was a bit confusing to perform a review about all noble metal NPs in Section 3. If the authors could consider to condense this section and move it to the Introduction?
(4) For Table 1 and 2, the significant digits of size and zeta-potential values should be unified.
(5) In the beginning of Section 4, it was suggested to firstly give a brief review of non-biological synthesis method of AgNPs, so as to compare with the biological methods.
(6) A summary upon the reaction mechanisms of AgNPs during biosynthesis would make the review more meaningful.
(7) In Section 5, the authors used a concept named ‘Green synthesis’. What was the relationship between biological synthesis and green synthesis? The concept should be used in consistence in a review.
(8) Was AgNO3 the only Ag source for AgNPs preparation?
(9) The authors were advised to discuss the industrialization translation of AgNPs in the Discussion Section.
(10) From Section 2 to 6, using subtitles would make the structure clearer.
(11) At the end of Conclusion, please add some concluding remarks about the future directions.
Reviewer 2 Report
Jorge L. Mejía-Méndez and co-authors present an overview of some biosynthetic noble-metal nanoparticles that were evaluated on cell line cultures. This review aimed to focus on lung cancer treatment and to summarize hypotheses about its mechanisms of action.
1. L.85 The sentence needs a reference.
2. In figure 1 text must be larger.
3. Tables 1 and 2 have too few entries, these are not enough for a review.
4. The authors stated that they consulted some databases "to present green synthesized AgNPs as a safe, strong, and cost-effective alternative to current treatment modalities of LC". However, references are about many types of cancer and not focused on lung cancer. Besides, just cytotoxic assays are described, and nothing about the cost efficiency of these alternatives is discussed.
- The title must be changed since the antineoplastic activity is not described but cytotoxic activity.
- The cost and efficiency of the biosynthetic noble metal-based nanoparticles compared to the first drug choice treatments need to be described.
- To present an objective review, the disadvantages of biosynthetic noble metal-based nanoparticles, such as reproducibility, degradation under physiological conditions, and photostability between others, must be explained.
Reviewer 3 Report
The following suggestions should be helpful to improve the quality of the manuscript.
1. The present form of the abstract is more generalized. It should be more specific as per the exact content of the manuscript.
2. The organization of the manuscript should be improved further to attract the reader's interest.
3. Some images should be incorporated to make the present form of the manuscript quickly understandable to the reader.
4. In my opinion, each section/subsection of the manuscript should also be discussed including some interesting illustrations.
5. In my opinion, Section 6 should be changed to contemporary research related to "Antineoplastic Activity Against Lung Cancer of Biosynthesized Silver Nanoparticles" with the incorporation of some interesting illustrations. A summary table should be included highlighting discussed contemporary research.
Author Response
Please the attachment.

Round 2
Reviewer 1 Report
The authors responded well to my questions.
Reviewer 2 Report
The authors correctly addressed the observations. I consider that the manuscript is ready to be published.
Reviewer 3 Report
The revised manuscript improved well after the inclusion of suggested comments. In my opinion, the present form of the manuscript should be considered for publication in Biomedicines.